# Observation of two types of charge-density-wave orders in superconducting $La_{2-x}Sr_xCuO_4$

J.-J. Wen [1,6], H. Huang [1,6], S.-J. Lee [1,6], H. Jang [1,2], J. Knight[1], Y.S. Lee[1,3], M. Fujita[4], K.M. Suzuki[4], S. Asano[4], S.A. Kivelson[5], C.-C. Kao[1] & J.-S. Lee [1]

The discovery of charge- and spin-density-wave (CDW/SDW) orders in superconducting cuprates has altered our perspective on the nature of high-temperature superconductivity (SC). However, it has proven difficult to fully elucidate the relationship between the density wave orders and SC. Here, using resonant soft X-ray scattering, we study the archetypal cuprate $La_{2-x}Sr_xCuO_4$ (LSCO) over a broad doping range. We reveal the existence of two types of CDW orders in LSCO, namely CDW stripe order and CDW short-range order (SRO). While the CDW-SRO is suppressed by SC, it is partially transformed into the CDW stripe order with developing SDW stripe order near the superconducting $T_c$. These findings indicate that the stripe orders and SC are inhomogeneously distributed in the superconducting $CuO_2$ planes of LSCO. This further suggests a new perspective on the putative pair-density-wave order that coexists with SC, SDW, and CDW orders.

[1] SLAC National Accelerator Laboratory, Menlo Park, California 94025, USA. [2] PAL-XFEL, Pohang Accelerator Laboratory, Gyeongbuk 37673, South Korea. [3] Department of Applied Physics, Stanford University, Stanford, CA 94305, USA. [4] Institute for Materials Research, Tohoku University, Sendai 980-8577, Japan. [5] Departments of Physics, Stanford University, Stanford, CA 94305, USA. [6]These authors contributed equally: J.-J. Wen, H. Huang, S.-J. Lee Correspondence and requests for materials should be addressed to J.-S.L. (email: jslee@slac.stanford.edu)

Since the 1986 discovery[1] of high-$T_c$ superconductivity (SC) in the cuprates, there has been an intense focus on understanding the essential physics of high-temperature SC. A major difficulty arises from the remarkably complex phase diagram. For example, in cuprates there are the so-called pseudogap and strange-metal regimes, and a variety of non-superconducting electronic orders (e.g., density-waves) associated with broken-symmetries[2–4]. Density wave order can occur on its own, or can coexist with superconducting order[5–18]. These complexities raise a fundamental question that has yet to be fully addressed—how do these orders/phases interact with SC[19–21]? Regarding this question, the role of charge-density-wave (CDW) order, which has been observed universally across different families of cuprates[10–18], is seemingly the most straightforward to interpret. For instance, in YBa$_2$Cu$_3$O$_{6+x}$ (YBCO) the CDW order, which develops above the superconducting $T_c$, is suppressed with the emergence of SC upon cooling[12,13]. Conversely, CDW order below $T_c$ is reinforced upon the suppression of the superconducting state by a magnetic field[22,23]. These phenomena clearly demonstrate a competitive relationship between CDW and SC. Another universally observed electronic order is spin-density-wave (SDW) order[5–8], but its relationship with CDW and SC is less clear-cut.

An ideal candidate-material for this study should contain regions in the phase diagram where CDW, SDW, and SC coexist. In YBCO, one of the most-studied cuprates, the experimentally observed CDW and SDW orders appear to be mutually incommensurate, and there is little or no regime in its phase diagram in which these three orders coexist[24,25]. From a phenomenological perspective, this can be accounted for in terms of a strongly repulsive bi-quadratic coupling in Landau-theory[26,27]. On the other hand, in the La-based cuprates, La$_{2-x-y}$(Ba,Sr)$_x$(Nd,Eu)$_y$-CuO$_4$, the CDW and SDW orders tend to satisfy a mutual commensurability condition on the ordering vectors, $q_{cdw} = 2q_{sdw}$[28], which implies the importance of a special tri-linear term in the Landau theory[26,27]. Moreover, there is generally a range of doping in which all three orders are observed to coexist[29–32]. In La$_{2-x}$Ba$_x$CuO$_4$ (LBCO), La$_{1.8-x}$Eu$_{0.2}$Sr$_x$CuO$_4$ (LESCO), and La$_{1.6-x}$Nd$_{0.4}$Sr$_x$CuO$_4$ (LNSCO), however, the phase diagram is further complicated by the existence of a low temperature

tetragonal (LTT) phase that tends to stabilize the CDW and SDW orders, and depress SC[28,33].

In this regard, we view La$_{2-x}$Sr$_x$CuO$_4$ (LSCO), which does not undergo LTT transition, as the ideal platform for the present study. As summarized in the schematic phase diagram in Fig. 1, in this study we find a temperature ($T$) and doping ($x$) dependence of the CDW order that differs in significant ways from what has been previously conjectured. We reveal distinct behaviors in various ranges of doping: (i) For $x < x_{cdw} \sim 0.1$ we observe no feature that can readily be identified with CDW order, although neutron scattering and NMR studies reveal the existence of stripe-like SDW order[9,34,35]. (ii) For $x_{cdw} \leq x \leq x_{sdw} \sim 0.135$ we observe CDW order with long correlation length that grows still longer even below superconducting $T_c$; from previous studies, $x_{sdw}$ is identified as the upper boundary of the regime in which (quasi) static SDW order persists at low $T$ (with an onset in the neighborhood of $T_c$);[9] we identify the growth of CDW correlations below $T_c$ with its mutual commensuration with the SDW order. (iii) For $x_{sdw} < x < x^\star \sim 0.18$, where neutron scattering indicates the existence of a spin-gap at low $T$ and NMR studies show no signatures of quasi-static magnetic order[9,35], we still observe clear evidence of well-developed short-range CDW correlations, but these are significantly suppressed at temperatures below $T_c$. (iv) For $x \geq x^\star$, no clear evidence of CDW or SDW order has been observed. In the following sections, detailed experimental findings and their implications will be discussed.

## Results

**Extending CDW phase diagram of LSCO.** The CDW order in LSCO has been observed previously by using X-ray scattering within a limited doping range in the underdoped side ($x = 0.11$, 0.12, and 0.13, also see Fig. 1)[31,36–38]. From a detailed analysis of the thermal evolution of the Seebeck coefficient, it was suggested that CDW order is confined to $x$ smaller than a critical doping $\sim 0.15$[39]. In this context, we first aimed to complete the CDW phase diagram in a wide doping range that extends from the underdoped to the overdoped side ($x = 0.075$, 0.10, 0.115, 0.12, 0.13, 0.144, 0.16, and 0.18). For this purpose, we employed a novel resonant soft X-ray scattering (RSXS) approach that significantly mitigates fluorescence background (see "Methods" section). Figure 2a shows a schematic of how the fluorescence rejection works during the RSXS measurement. A large area detector was used to measure both the signal of interest (near CDW area) and the background signal (away from CDW) simultaneously. By subtracting the background (Fig. 2b, c), we achieve a significantly improved detecting sensitivity in the RSXS, allowing us to explore weak CDW signals. Figure 2d shows the scattering intensity maps along the $h$-/$k$-direction centered at $q_{cdw} \sim (-0.23, 0, l)$ r.l.u., which were measured at respective $T_c$. Clear CDW peaks are observed for $0.1 < x < 0.18$ ($x = 0.115$, 0.12, 0.13, 0.144, and 0.16). Furthermore, in the low dopings we can clearly resolve a CDW peak splitting along the $k$-direction, which is consistent with previous reports of CDW and SDW coexistence in $x = 0.12$[37,40]. This splitting becomes rather unclear for $x \geq 0.144$, a behavior that seems to be correlated with the disappearance of SDW stripe order above $x_{sdw} \sim 0.135$[9].

**Two types of CDW orders.** As a next step, we explore the $T$ dependence of the CDW orders. Figures 3a, b show the projected CDW signals along the $h$-direction for $x = 0.144$ and 0.13 LSCO, respectively (See Supplementary Fig. 1 for corresponding data for $x = 0.115$, 0.12, and 0.16). Upon cooling, the CDW peak in $x = 0.144$ ($x > x_{sdw}$) increases, reaching the maximum around $T_c$, and then decreases as SC emerges below $T_c$. This is consistent with the expectation that CDW competes with SC[12,41,42]. For a slightly

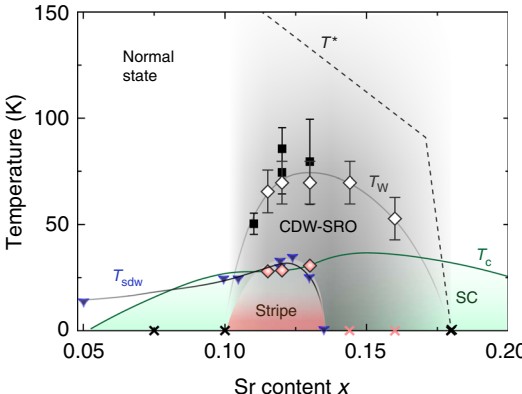

**Fig. 1** Phase diagram of LSCO. A sketch of the LSCO phase diagram. The open and filled diamonds denote $T_w$ and the onset temperature of CDW stripe order determined in this study, respectively. The error bar of $T_w$ is estimated to be 10 K. The filled squares are the CDW onset temperature reported in previous X-ray scattering studies[31,39]. $T_{sdw}$ is the SDW onset temperature determined from neutron scattering measurements[9,39]. $T_c$ is the superconducting transition temperature[39] and $T^\star$ is the pseudogap temperature[43]. The black and red cross symbols indicate the doping levels where no CDW-SRO or CDW stripe order have been detected in this study, respectively

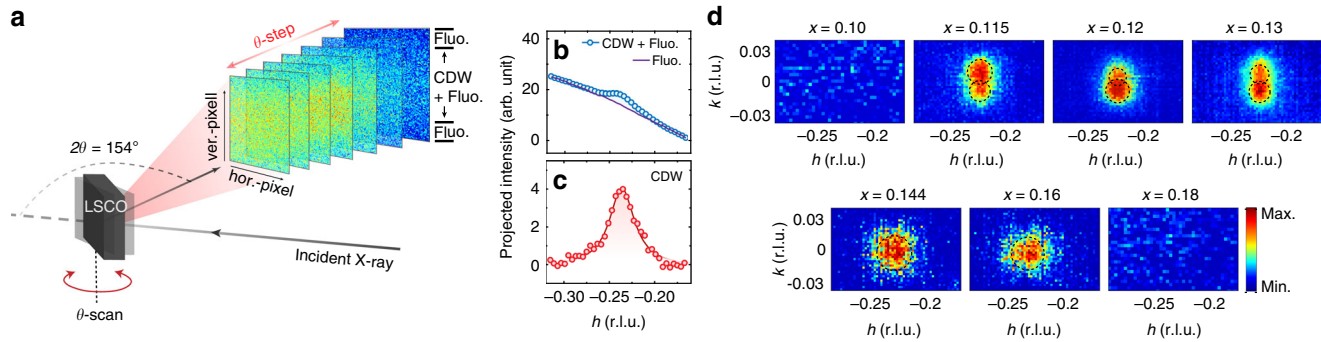

**Fig. 2** RSXS measurements on LSCO. **a** A schematic sketch showing the RSXS experimental setup. The top and bottom parts in each CCD image are regarded as the fluorescence (Fluo.) background. **b** The projected intensity profiles along the $h$-direction at both the CDW area ($k \sim 0$ r.l.u.) and fluorescence area ($k \sim \pm 0.04$ r.l.u.). **c** The CDW profile after subtracting the fluorescence background. The data in (**b**) and (**c**) is for $x = 0.13$ sample measured at 23 K. **d** Scattering patterns for various LSCO samples after subtracting the background. Measurements were taken at respective $T_c$. The dashed circles outline the intensity contour

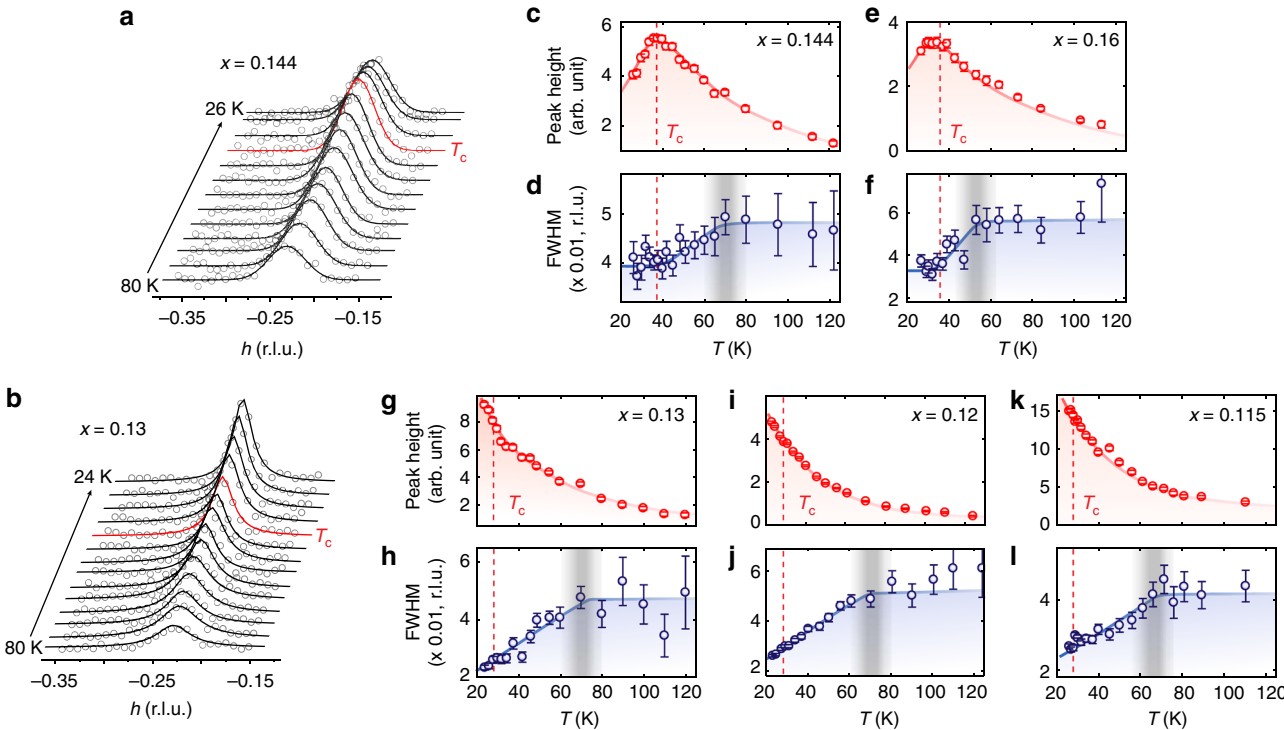

**Fig. 3** Temperature dependence of the CDW order in LSCO. **a**, **b** Projected scattering profiles along $h$ as a function of temperature for $x = 0.144$ (**a**) and 0.13 (**b**) LSCO. The solid lines are Lorentzian fits. The red curves denote the fits for data at $T_c$. **c–l** Temperature dependent CDW peak heights and full-width at half-maximum (FWHM) extracted from the fits for various LSCO samples. The red dashed lines and vertical gray shades denote $T_c$ and $T_w$, respectively. The colored shades and lines are guides-to-the-eye. The error bars represent 1 standard deviation (s.d.) of the fit parameters. Source data are provided as a Source Data file

less doped sample of $x = 0.13$ ($x < x_{sdw}$), while the high-$T$ behavior is quite similar, the evolution of the CDW correlations through $T_c$ is surprisingly different. The CDW peak continues to increase with decreasing temperature even below $T_c$, which has not been unambiguously reported in the previous studies on LSCO[31,36–38] (Supplementary Discussion and Supplementary Fig. 2).

For quantitative investigation of the $T$ dependence, we fitted the CDW peaks with the Lorentzian function (Fig. 3c–l). For temperatures above $T_c$, the CDW correlations in all dopings follow similar trend. The peak intensities in all dopings decrease continuously with increasing temperature, showing no clear indication of an onset-$T$. Notably, the CDW peak intensity in our

study persists even above recent estimates of the pseudogap temperature ($T^*$) (see Supplementary Fig. 3)[43]. Instead of an onset-$T$, we could extract a characteristic temperature ($T_w$) from the $T$-dependent full width at half maximum (FWHM) plots. The $T_w$ is the temperature above which the CDW correlation length $\xi_{cdw}$ (calculated as 2/FWHM) becomes approximately $T$-independent and below which the $\xi_{cdw}$ increases with decreasing temperature. Such extracted $T_w$ matches previously reported CDW intensity onset-$T$ as shown in Fig. 1. The $T$-independent $\xi_{cdw}$ at high temperatures is reasonably short at $\sim 24$ Å, but is still longer than the CDW wave-length $\lambda_{cdw} \sim 4a \sim 15$ Å. Note that we view $\xi_{cdw} \geq \lambda_{cdw}$ or equivalently FWHM $\leq q_{cdw}/\pi$, as a minimal condition for unambiguously identifying a diffraction peak as

indicative of CDW order. For temperatures below $T_c$, clear contrast emerges. For $x > x_{sdw}$ (Fig. 3c–f), the increasing trend in the $\xi_{cdw}$ stops or even slightly reverses below $T_c$, which is coincident with the suppression of the CDW peak intensity. This is similar to what is commonly observed in the superconducting YBCO and can be readily explained by the competition between the CDW and the SC[12,41,42]. Hereinafter, we refer to this as CDW short-range order (SRO). In stark contrast, for $x < x_{sdw}$ (Fig. 3g–l), both $\xi_{cdw}$ and peak intensity keep increasing below $T_c$. Such contrast cannot be attributed in any naïve way to enhanced disorder-induced pinning of the CDW-SRO in $x \le x_{sdw}$ samples (i.e., $x = 0.115$, 0.12, and 0.13). Given that disorder in LSCO is mainly induced by Sr substitution, these less doped samples are expected, if anything, to have less pinning. This indicates that the enhanced CDW at low-$T$ is a new type of CDW order.

**Interplay between the two types of CDW and SDW.** To scrutinize this new type of CDW order and the CDW-SRO, we replotted CDW intensities vs. $T/T_c$ for different doping levels with the intensities at $T_c$ normalized to unity (see Fig. 4a). This figure clearly shows two drastically different temperature regimes separated by $T/T_c = 1$. For $T < T_c$, the contrasting behavior of the CDW intensity between $x < x_{sdw}$ ($x = 0.115$, 0.12, and 0.13, where static SDW order develops below $\sim T_c$) and $x > x_{sdw}$ ($x = 0.144$ and 0.16, where static SDW order is absent) indicates that the enhancement of the CDW order is associated with the development of SDW order. This is further supported by the step-like increase of the low-$T$ CDW correlation length for $x < x_{sdw}$ as shown in Fig. 4b. Furthermore, the CDW and the SDW in LSCO are found to follow the $q_{cdw} \sim 2q_{sdw}$ relation (see Supplementary Fig. 4), which is analogous to the prototypical stripe order in LBCO[30]. We therefore identify the new type of CDW as the CDW stripe order. For $T > T_c$, the normalized CDW intensities for all doping levels track each other, suggesting that the CDW orders above $T_c$ are of the same short-range-order type. These observations, in conjunction with the fact that neither of the two CDW types exists at doping levels $x = 0.1$ and below even though clear SDW stripe order exists in these dopings[9,34], imply that the CDW stripe order is not simply parasitic to the SDW stripe order, but rather is due to a cooperative interaction between SDW and the preexisting CDW-SRO. In short, these findings provide clear and direct evidence of the intertwining between CDW and SDW orders in LSCO.

## Discussion

We now discuss the implications of our results on SC. Previous NMR, μSR and neutron studies of the magnetic correlations for $x < x_{sdw}$ have demonstrated that the static SDW order is inhomogeneously distributed in the superconducting $CuO_2$ plane at $T < T_c$[9,35,44,45]. The linkage between the CDW stripe order and the growth of the SDW order for $x_{cdw} < x < x_{sdw}$ and $T < T_c$ implies that they are coincident in the same regions. On the other hand, the CDW-SRO, which is presumably uniformly distributed at high-$T$, is likely to continue to persist in the rest of the regions where SDW order is absent at low-$T$, but is suppressed by the development of SC. We illustrate such low-$T$ state in Fig. 5, which represents a $CuO_2$ plane consisting of regions of uniform $d$-wave SC and regions with the stripe orders. This is the sort of structure expected when two-phase coexistence is frustrated either by disorder or long-range interactions[3,4], and is consistent with recent numerical studies that find near degeneracy between SC and stripe state[46,47]. This picture also provides a plausible explanation for the seemingly contradictory CDW results on $x = 0.12$ LSCO, where the CDW peak intensity increases (ref. [36] and this study), decreases[31], or levels off[37,38] upon cooling below $T_c$. Such

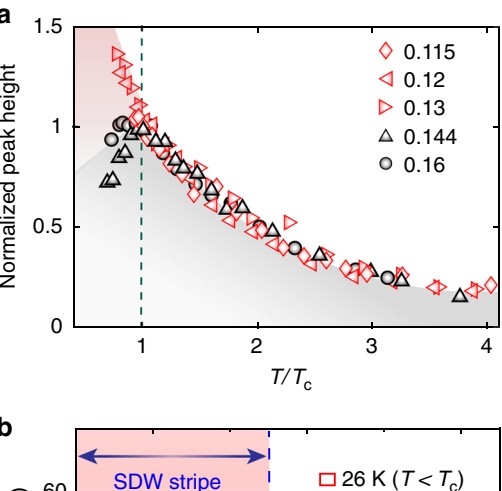

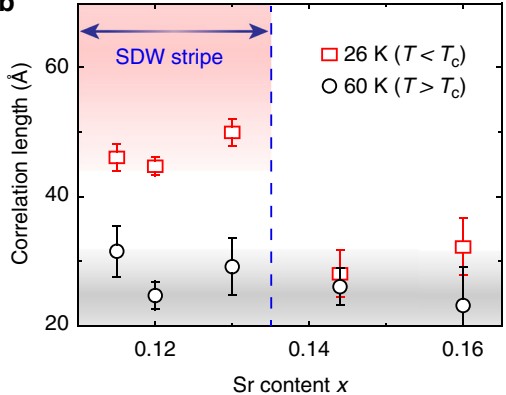

**Fig. 4** Comparison of CDW correlations with and without the SDW stripe order in LSCO. **a** Temperature dependent CDW peak heights for $x < x_{sdw}$ (red) and $x > x_{sdw}$ (black). Both the peak heights and temperatures are normalized to the values at respective $T_c$. **b** Doping dependent in-plane CDW correlation length for LSCO samples at 26 K and 60 K. The dashed line denotes the $x_{sdw}$. The colored shades are guides-to-the-eye. The error bars represent 1 s.d

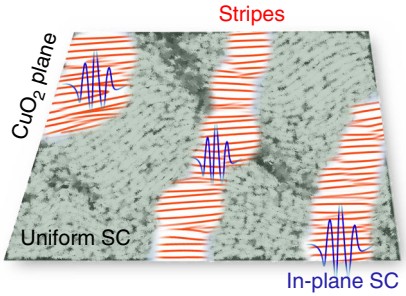

**Fig. 5** Inhomogeneous electronic orders in LSCO. Artistic illustration of the stripe-ordered $CuO_2$ plane in LSCO. The green colored area denotes the uniform SC state. The distorted big-waves illustration indicates the weakened CDW-SRO. Red colored pattern illustrates the stripe orders. The blue modulations formed around the stripe-ordered areas depict the putative PDW

discrepancy can be due to the different volume fractions of CDW stripe order that were probed in different experiments (see "Supplementary Discussion" and Supplementary Fig. 2).

The fact that SDW/CDW stripe order, and SC order all onset at roughly the same $T$, and that the SC $T_c$ remains sharp implies that these orders are intimately related. In particular, these findings suggest that substantial SC order permeates the stripe-ordered

regions of LSCO. Given that where such stripe order occurs in LBCO, it has been proposed based on transport anomalies that the SC order primarily takes the form of the pair-density-wave (PDW)[48,49], it is natural to conjecture that the same is true of the stripe-ordered regions in LSCO. If this is the case, altering the balance between stripe (PDW) ordered and uniform SC regions in $CuO_2$ plane is expected to affect $c$-axis SC coherence much more dramatically than in-plane SC properties. Indeed, our conjecture is supported by the observation that for $x \sim 0.1$ LSCO, an applied magnetic field much smaller than $H_{c2}$ strongly enhances stripe order[50] and abruptly quenches $c$-axis SC coherence measured optically[51]. Thus the connection between stripe orders and PDW is likely universal in superconducting cuprates beyond LBCO[48,49]. Future experimental work using a spatially-resolved probe, such as STM, will provide further evidence regarding the spatial distribution of SC order in LSCO. Finally, turning to high $T$ and $x$, the observation that CDW-SRO persists above $T^*$, without a change in the correlation length, opens the possibility that it might be directly correlated with other pseudogap phenomena. At minimum, the upper critical doping boundary of the CDW-SRO is strikingly close to the pseudogap critical doping, $x^* = 0.18$[39,43]. Similar correlation has also been noticed in a previous $Bi_2Sr_2CaCu_2O_{8+\delta}$ study[52].

## Methods

**Sample preparation.** High-quality single crystals of LSCO with nominal concentrations $x = 0.075$, 0.10, 0.115, 0.12, 0.13, 0.144, 0.16, and 0.18 were grown by the traveling solvent floating zone method. The typical growth rate was 1.0 mm h$^{-1}$ and a 50–60 mm-long crystal rod was successfully obtained for each concentration. A 10 mm-long crystalline piece from the end part of each grown rod was annealed in oxygen gas flow to minimize oxygen deficiencies. Before the RSXS measurements, using a Quantum Design PPMS we characterized their superconducting $T_c$ as the mid-point of the transition. Such obtained $T_c$ for LSCO samples, $x = 0.075$, 0.10, 0.115, 0.12, 0.13, 0.144, 0.16, and 0.18, are 15.0(2), 27.4(2), 27.3(2), 28.4(2), 30.8(2), 37.5(2), 35.5(2), and 30.4(2) K, respectively as summarized in Supplementary Fig. 5.

For the RSXS measurements, we prepared all the samples with a typical dimension of 1.5 mm × 1.5 mm × 2.5 mm ($a \times b \times c$ axis). To achieve a fresh $c$-axis normal surface, each sample was ex situ cleaved, before transported into the ultra-high vacuum chamber (base pressure = $8 \times 10^{-10}$ Torr).

**RSXS measurement.** All the experiments were carried out at beamline 13–3 of the Stanford Synchrotron Radiation Lightsouce (SSRL). The sample was mounted on an in-vacuum 4-circle diffractometer. The sample temperature was controlled by an open-circle helium cryostat. Incident photon polarization was fixed as sigma (vertical linear) polarization. Exact ($h$, 0, $l$) scattering plane was aligned by the measured (0, 0, 2), (−1, 0, 1), and (1, 0, 1) structural Bragg reflections at the photon energy ~1770 eV. The energy of the Cu $L_3$-edge was determined by X-ray absorption spectroscopy (see Supplementary Fig. 6). The typical energy resolution of incident X-ray at the Cu $L$-edge region is 0.1 eV.

A 256 × 1024 pixel CCD detector was used for the RSXS measurements. Each pixel is of size 26 μm × 26 μm. To minimize the geometric effect in the measurements (the background slope as a function of $\theta$-angle), the vertically wide 2D-CCD detector was fixed at $2\theta \sim 154°$ which is the highest achievable $2\theta$ position without blocking incoming X-rays. By rotating the sample (i.e., $\theta$-scan) with 0.5° per step, we obtained the $h$-dependence. In a typical CDW measurement, each CCD image was accumulated with an exposure time of 3–5 s at each $\theta$. A beam shutter was used to cut the incoming X-ray beam between two consecutive CCD shots to prevent undesired collection of X-ray photons during read-out. In front of the CCD is mounted a 100 nm Parylene/100 nm Al filter to stop electrons emitted from the sample from contributing to CCD signals, ensuring our CCD measures purely X-ray photons. During the $\theta$-scan, each CCD image covers scattering intensities from the well-aligned ($h$, 0, $l$) scattering plane near the detector center as well as from off-scattering planes ($h$, ±$k$, $l$) at the top/bottom area of the CCD. As described in the main text (Fig. 2a), the scattering intensity near the center of the detector corresponds to the signal of interest (i.e., CDW in this case), while the signals at the off-centered areas are dominated by the fluorescence background. This simultaneously recorded background signal (~256 × 50 pixels near the top and bottom of the CCD detector respectively) was used to subtract out the zeroth order fluorescence contribution near the CDW area. These background regions correspond to $k \sim \pm 0.04$ r.l.u., which is considerably far away from the center of the CDW peak, considering the finite width of the CDW peak.

For data analysis, each CCD pixel was converted to an $hkl$ reciprocal space index, and the resultant three-dimensional scattering intensity data set was projected onto different planes/directions for subsequent analysis.

## Data availability

The data that support the findings of this study are available from the corresponding author upon reasonable request.

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

## Acknowledgements

We thank John M. Tranquada, Mark P. M. Dean, Stephen Hayden, and J. C. Séamus Davis for insightful discussions. All soft X-ray experiments were carried out at the SSRL (beamline 13–3), SLAC National Accelerator Laboratory, supported by the U.S. Department of Energy, Office of Science, Office of Basic Energy Sciences under Contract No. DE-AC02-76SF00515. J.W. and Y.S.L. acknowledges the support by the Department of Energy, Office of Science, Basic Energy Sciences, Materials Sciences and Engineering Division, under contract DE-AC02-76SF00515. S.A.K. acknowledges the support by Department of Energy, Office of Basic Energy Sciences, under contract no. DE-AC02-76SF00515 at Stanford. M.F. is supported by Grant-in-Aid for Scientific Research (A) (Grant No. 16H02125) and Scientific Research (C) (Grant No. 16K05460).

## Author contributions

J.W., H.H., S.L., H.J., J.K., and J.-S.L. carried out the RSXS experiment and analyzed the data. M.F., K.S., and S.A. synthesized the materials. J.W., H.H., S.L., Y.S.L., F.M., S.A.K., C.-C.K., and J.-S.L. wrote the manuscript with input from all authors. J.-S.L. coordinated the project.

## Additional information

**Competing interests:** The authors declare no competing interests.

