## [Peer Review File · Nature Communications]

Reviewers' comments:

Reviewer #1 (Remarks to the Author):

The manuscript by Wen et al. reports on the observation of two forms of charge order in LSCO cuprate compounds: one manifesting itself as a charge-density-wave similar to what has been reported in recent years in most cuprate compounds; and the other one displaying the characteristics of stripe order.

From a technical standpoint, this study has been performed to very high scientific standards and provides a rather comprehensive investigation of the doping and temperature dependence of translational-symmetry breaking orders cohabiting in the underdoped region of the phase diagram of LSCO. This compound is indeed an almost ideal playground to study the interplay between CDWs, stripes, and superconductivity, and the observations reported here are very significant in relation to the description and understanding of the physics of collective electronic orders in cuprates.

The main conclusions of this study appear to be sufficiently well-supported by the experimental data, although I have a few related remarks below that I invite the authors to consider in their revisions. I hope the authors can address my points, in which case I will be happy to give my best endorsement for publication of this nice study and manuscript in Nature Communications.

Here are some specific remarks:

- The authors conclude that the enhancement of CDW below T_c (such enhancement is not so obvious, by the way, at least judging from the temperature curves) reveals the presence of a different type of charge order which is here linked to the known stripe phenomenon. While I agree that the varying temperature dependence of the scattering signal is likely indicative of two forms of charge order, it is not clear to me what is the evidence in support of the second kind (stripe) having an onset at around T_c . What part of the body of experimental evidence unambiguously suggests that there is a second entity rising above the first one (CDW-SRO) below a certain temperature? The absence of any clear discontinuity in the temperature- or doping-dependence of the scattering signatures of charge order makes this hard to appreciate for me, and I hope the authors can elucidate this aspect.
- The authors also add that "these findings suggest that substantial SC order permeates the stripe-ordered regions of LSCO". What aspects of the experimental data is in support of this conclusion? I would have expected that a statement of this kind would require a spatially-resolved probe like STM. I hope the authors can clarify this passage.
- In the abstract, the authors characterize their experimental approach as "novel". With all due respect to the experimental execution and data analysis, which are bulletproof, I wonder if the procedure of subtracting the background from a side region of the photon detector is sufficient to claim invention of a novel approach. On top of that, my understanding is that the fluorescence signal depends on the probing geometry and on the photon incidence and takeoff angles, therefore the estimate of the background is not exact (in the sense that it is not evaluated at the angles where the CDW peak is present, but rather off from it). In any case, I am happy to reconsider my position above if the authors can elaborate further on the elements of novelty of these measurements.
- In reading the manuscript, I was surprised that there is hardly any discussion on the ordering vectors, which appear to be nearly independent on doping. Is the CDW-stripe ordering vector related to the SDW ordering vector (for the same doping levels) by a factor 2, as expected in the stripe phase?

Reviewer #2 (Remarks to the Author):

the paper by Wen et al. provides a systematic doping and temperature dependence of charge density waves in La-based cuprates. Its main results are i) the observation of the CDW on a broader doping range than previously reported, and ii) the existence of two types of CDW, depending on whether it is associated with the formation of a SDW below the superconducting T_c .

This is an interesting study that provides fresh insights on the interplay between spin and charge degrees of freedom (and thus on that between charge density waves and stripes) in the underdoped cuprates, a most interesting and highly debated issue. The paper is well presented and convincingly written and, in my opinion, meets all the standards for publication in Nature Communications.

I initially felt that some aspects of the discussion had been somewhat overlooked, in particular regarding previous work on LSCO compounds. Indeed, the present paper reports on data that are seemingly contradicting some of the previous publications on the subject, for instance as far as the temperature dependence of the CDW close to the $1/8$ doping is concerned: refs. 38 and 31 both clearly show a decreasing intensity of the CDW peak below T_c . Refs. 36 and 37 are less clear in this respect. In the present paper, the authors write 'The CDW peak continues to increase with decreasing temperature even below T_c , which has not been unambiguously reported ...'

At first, I found this statement misleading, but then realized this was satisfactorily discussed in the supplementary information (fig. S5). It should then just be mentioned in the main text that this point is discussed in the supplementary information. Same goes with the doping and temperature dependence of the incommensurability.

These two points were my main concerns, and after realizing that they were adequately discussed in the supplementary, I cannot find anything holding me back from recommending this paper for publication in Nature Communications.

Reviewer #3 (Remarks to the Author):

The manuscript by Wen et al. describes a clearly outlined, systematic resonant elastic x-ray scattering study of the CDW order in LSCO as a function of doping and temperature. Through elimination of the major fluorescence background in their REXS data the authors were able to more precisely analyse the CDW peak intensity, width and its temperature dependence over a wider doping range in LSCO as compared to previous studies. Their main finding is a distinctively larger CDW correlation length in the doping range of coexisting SDW compared to samples with higher doping levels. Furthermore, the authors observe a short-range CDW diffraction peak at higher temperatures, even higher than the pseudogap temperature (without further change in correlation length) which seems to compete with SC for the higher doping levels as observed previously. Technically, I have no objections. The data quality and analysis appears good and the observed changes in FWHM of the CDW peaks are significant (about 0.02 rlu) and clearly above the error bar. One of the main messages is that the CDW stripe order originates from cooperative interaction between the SDW and pre-existing short-range CDW order in the underdoped region. The authors suggest that in analogy to LBCO a PDW form of the SC order parameter is present in the stripe ordered regions. This argument I find a bit weak and indirect.

Overall, I think the data presented is of high quality and interest. The manuscript is clearly written and sufficiently based on previous studies. In my opinion, the presented data provide new valuable insights to understand the complex phase diagrams of cuprates. Therefore, I recommend the manuscript for publication in Nature Communication.

Response to the Referees' comments

We appreciate all three Referees for their careful read-throughs and positive evaluations of the manuscript. We also thank them for supporting publication of our manuscript in *Nature Communications* and for their important and valuable comments. We have made corresponding changes to the manuscript that we believe make it clearer and more complete. Below, we present our point-by-point responses to referees' queries. With our response and the changes in the revised manuscript we believe that we have addressed these requests and hope that the manuscript is now suitable for publication in *Nature Communications*.

Reply to Referee #1:

The authors conclude that the enhancement of CDW below T_c (such enhancement is not so obvious, by the way, at least judging from the temperature curves) reveals the presence of a different type of charge order which is here linked to the known stripe phenomenon. While I agree that the varying temperature dependence of the scattering signal is likely indicative of two forms of charge order, it is not clear to me what is the evidence in support of the second kind (stripe) having an onset at around T_c . What part of the body of experimental evidence unambiguously suggests that there is a second entity rising above the first one (CDW-SRO) below a certain temperature? The absence of any clear discontinuity in the temperature- or doping-dependence of the scattering signatures of charge order makes this hard to appreciate for me, and I hope the authors can elucidate this aspect.

→ First, we appreciate the referee for the evaluation on the high quality of our paper. Also, thanks for bringing our attention to this part. We certainly agree that it is difficult to infer that CDW stripe order onsets $\sim T_c$ by only looking at the CDW temperature dependence of individual sample. As the referee pointed out, on the other hand, this becomes possible through the comparison of CDW temperature dependences between different doping levels – *This approach is exactly how we were able to identify two forms of CDW orders.*

For more details, as shown in Fig. 4A and corresponding text, the CDW temperature dependences are the same for $T > T_c$ regardless of the doping level. We therefore identify the CDW for $T > T_c$ to be of the same CDW-SRO nature. On the other hand, for $T < T_c$, the CDW intensities for doping levels $x < x_{\text{sdw}}$ keeps increasing with decreasing temperature. This is not what is expected for CDW-SRO, which is suppressed below T_c . As shown in Fig. 4A, this enhancement with respect to CDW-SRO occurs below T_c . Combining this observation together with previous neutron scattering measurements that found SDW stripe order to onset $\sim T_c$, we deduce T_c to be also the onset temperature for CDW stripe order. Admittedly there does not appear to be clear discontinuity or anomaly at the onset temperature. One possible reason is that random disorder, which inevitably exists in LSCO due to Sr substitution, precludes true long-range-order of the CDW stripe order [See e.g. Phys. Rev. B 92, 174505 (2015)]. Indeed, the correlation length of the CDW stripe order (See Fig. 4B) remains finite down to low temperatures. Since this system is not able to achieve long range CDW stripe order, only a smooth development of CDW stripe order is observed below the onset temperature, instead of singular behaviors that are usually expected when long-range-order onsets.

The authors also add that “these findings suggest that substantial SC order permeates the stripe-ordered regions of LSCO”. What aspects of the experimental data is in support of this conclusion? I would have

expected that a statement of this kind would require a spatially-resolved probe like STM. I hope the authors can clarify this passage.

→ Thanks for bringing our attention to this point. We agree with the referee that in our RSXS experiment we do not have direct evidence regarding the spatial distribution of the superconducting order. However, as argued in the main text, several indirect evidences suggest that there can be substantial spatial overlap between stripe orders and SC correlations. First, the observation that SDW stripe order, CDW stripe order, and bulk SC all onset $\sim T_c$ indicate these orders are closely related to each other. Moreover, since the bulk superconducting transition remains sharp despite simultaneous development of the stripe orders, it is not unreasonable to expect that some form of SC correlation, likely a spatially modulated SC correlation in this case, exists in the stripe-ordered regions. Second, in $\text{La}_{2-x}\text{Ba}_x\text{CuO}_4$, where the interplay between stripe orders and SC has been extensively studied, there are evidences including transport measurements [e.g. Phys. Rev. Lett. 99, 067001 (2007)] that spatially modulated SC correlations in the form of pair-density-wave coexist with the stripe order. Based on the similarity between LSCO and LBCO, it is reasonable to conjecture that the same scenario could also occur in the striped-ordered regions in LSCO. Finally, in LSCO a moderate magnetic field (much smaller than H_{c2}) is found to greatly enhance SDW order at the expense of inter-layer superconducting coherence [Nature 415, 299 (2002), Phys. Rev. Lett. 104, 157002 (2010)]. Considering that pair-density-wave frustrates inter-layer Josephson coupling, this is consistent with the scenario that pair-density-wave coexist with stripe order and both are enhanced with magnetic field. These indirect evidences provide support that SC order permeates the stripe-ordered regions in LSCO.

Nevertheless, we appreciate the referee's point. For clarification, we have added "... Future experimental work using a spatially-resolved probe such as STM will provide further evidence regarding the spatial distribution of SC order in LSCO." in the last paragraph of the revised manuscript.

In the abstract, the authors characterize their experimental approach as "novel". With all due respect to the experimental execution and data analysis, which are bulletproof, I wonder if the procedure of subtracting the background from a side region of the photon detector is sufficient to claim invention of a novel approach. On top of that, my understanding is that the fluorescence signal depends on the probing geometry and on the photon incidence and takeoff angles, therefore the estimate of the background is not exact (in the sense that it is not evaluated at the angles where the CDW peak is present, but rather off from it). In any case, I am happy to reconsider my position above if the authors can elaborate further on the elements of novelty of these measurements.

→ As the referee pointed out, the main improvement of our RSXS procedure is to use a part of a wide 2-dimensional (2D) detector to evaluate the fluorescent background. Compared with our experiences of using a point detector and the 2D detector without this proper fluorescence subtraction, this new procedure significantly improves the ratio of signal to background. This improved sensitivity allows us to carry out this CDW study in the broad doping range. Nevertheless, we agree with the referee that in terms of experimental setup/operation, our new approach does not differ significantly from what is usually done in RSXS measurements. However, our detecting sensitivity is not achievable in the conventional RSXS procedure. Also, although our fluorescent background subtraction procedure might seem an obvious thing to do, we have not seen similar data treatment in the CDW literature. In this context, our original intent of using "novel" is to highlight the enhanced sensitivity of our approach, not to claim the invention of a new RSXS experimental method.

However, we also agree with the referee that our subtraction is not perfect, but it provides a satisfactory first order fluorescent correction to the data. To avoid any confusion, we have removed "novel" from the abstract in the revised manuscript.

In reading the manuscript, I was surprised that there is hardly any discussion on the ordering vectors, which appear to be nearly independent on doping. Is the CDW-stripe ordering vector related to the SDW ordering vector (for the same doping levels) by a factor 2, as expected in the stripe phase?

→ The discussion regarding the CDW ordering wave vector is presented in the Supplementary Fig.4. The SDW ordering wave vector is also only weakly doping-dependent in the doping range that we studied. And indeed, the $q_{\text{cdw}} \approx 2q_{\text{sdw}}$ relationship holds within experimental uncertainty.

Reply to referee #2:

... At first, I found this statement misleading, but then realized this was satisfactorily discussed in the supplementary information (fig. S5). It should then just be mentioned in the main text that this point is discussed in the supplementary information. Same goes with the doping and temperature dependence of the incommensurability. These two points were my main concerns, and after realizing that they were adequately discussed in the supplementary, I cannot find anything holding me back from recommending this paper for publication in Nature Communications.

→ We appreciate the referee for the thorough evaluation of our manuscript and the recommendation of publication of our work. In the revised manuscript, we have accordingly made more explicit references to the Supplementary information regarding the existing CDW work on LSCO and the doping dependence of the CDW wave vector.

Reply to referee #3:

Overall, I think the data presented is of high quality and interest. The manuscript is clearly written and sufficiently based on previous studies. In my opinion, the presented data provide new valuable insights to understand the complex phase diagrams of cuprates. Therefore, I recommend the manuscript for publication in Nature Communication.

→ We appreciate the referee for the thorough evaluation of our manuscript and the recommendation of publication of our work.

Summary of Changes:

1. We have added "... Future experimental work using a spatially-resolved probe such as STM will provide further evidence regarding the spatial distribution of SC order in LSCO." in the last paragraph.
2. In the abstract, we changed "Here using a novel resonant soft x-ray scattering approach..." to "Here using resonant soft x-ray scattering..."
3. We have added more explicit references to Supplementary Fig. 4 when discussing about doping dependence of CDW wave vector, and to Supplementary discussion and Supplementary Fig. 5 when discussing about existing CDW work on LSCO.
4. We have formatted both the main text and Supplementary information according to the *Nature Communications* format requirement.

REVIEWERS' COMMENTS:

Reviewer #1 (Remarks to the Author):

I have carefully examined the authors' response and the new manuscript version. I believe the revised manuscript and authors' notes address my previous comments in their entirety.